# ATHENA-SERVE: AN INTELLIGENT SCHEDULING LLM SERVING SYSTEM VIA HORIZON-COST PREDICTION AND HIERARCHICAL RL

## ABSTRACT

Online inference serving for large language models (LLMs) is foundational infrastructure for conversational agents, retrieval-augmented generation, and multi-tenant intelligent applications. Its core objective is to meet strict latency SLOs under heterogeneous and bursty workloads. However, existing systems suffer from bursty arrivals and long-tailed output lengths that drive peak cache pressure and bandwidth contention, as well as the brittleness of FCFS or shortest-job heuristics under noisy length regression and distribution shift—ultimately compounding tail-latency violations and head-of-line (HoL) blocking. We propose ATHENA-Serve, a deployable, horizon–cost–aware LLM serving scheduler. ATHENA-Serve converts predicted generation horizons into calibrated memory and compute budgets. Rather than forecasting exact trajectories, it senses each request's compute demands and memory footprint. Guided by these budgeted signals, ATHENA-Serve proactively constrains batching and concurrency to smooth memory peaks, while conditioning scheduling decisions on global system signals.

## 1 INTRODUCTION

Online inference serving for large language models (LLMs) underpins search, question answering, code assistance, and conversational systems Achiam et al. (2023); Touvron et al. (2023b;a); Brown et al. (2020); Radford et al. (2021). As deployments scale, bottlenecks shift from compute-bound kernels to queueing-centered resource coordination—driven by peak KV-cache footprint, bandwidth contention, and batching mismatches Qin et al. (2025); Gao et al. (2025); Dong et al. (2025). These interactions sharpen the throughput–latency trade-off, complicating responsiveness at high utilization. Addressing them is essential to sustain efficiency and scalability under tightening latency constraints.

Despite steady advances in low-level execution and memory management, the serving layer faces three persistent challenges: (i) *workload uncertainty and heterogeneity*—bursty, nonstationary arrivals induce sharp KV-cache peaks and severe decoding contention Xiang et al. (2025); Su et al. (2025); (ii) *brittleness of SJF/SRPT*—size-only policies overlook VRAM/KV pressure and batching/prefill–decode couplings, amplifying congestion, unfairness, and instability under bursty, multi-server workloads Bansal & Harchol-Balter (2001); Grosof et al. (2019); and (iii) *limited system- and horizon-awareness*—myopic optimization from instantaneous snapshots Xiao et al. (2018) fails to jointly balance latency, KV peaks, and fairness at high load, and to anticipate future footprints (e.g., KV-cache trajectories) required for tail suppression Dean & Barroso (2013).

Our objective is to build a serving framework that couples fine-grained observability of the current system state with forward-looking assessment of future load, and uses both to make interpretable, deployable scheduling and admission decisions Lee et al. (2025). In the present state, the scheduler tracks GPU compute and VRAM utilization, intra/inter-batch compute structure, and traffic burstiness, adapting actions accordingly Liang et al. (2024); Sheng et al. (2023): when compute is slack, it enlarges batches to raise parallelism; when VRAM pressure dominates, it prioritizes completing short outputs to promptly release bandwidth and KV-cache. In the future state, decisions go beyond greedy admit/evict tied to instantaneous memory watermarks: predicted output lengths—and

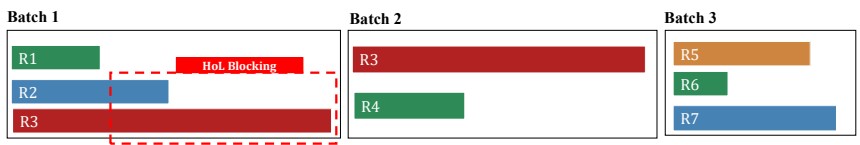

(a) vLLM (FCFS): Head-of-Line (HoL) Blocking under FIFO Batch Construction.

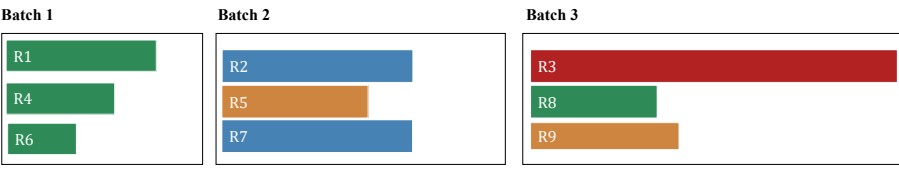

(b) ATHENA: HoL Mitigation via Horizon- and State-Aware Scheduling.

Figure 1: Batch-Timeline Illustration of Head-of-Line Dynamics and Mitigation. ATHENA-Serve alleviates strict FIFO constraints by converting horizon priors into budgeted, state-aware actions, selectively prioritizing short requests without causing starvation. This approach reduces decode-stage serialization and decreases time-to-first-token (TTFT).

the induced KV-cache and compute budgets—constrain global peaks, trigger admission limits, and proactively shape load when necessary Mao et al. (2016; 2019).

This work introduces ATHENA-SERVE, which operationalizes forward-looking, stateful control by fusing calibrated horizon–cost prediction with hierarchical policy optimization, producing interpretable admission, batching, and decoding-concurrency decisions at high utilization. The major contributions are summarized as follows:

- **Budget semantics via calibrated horizons.** ORACLE converts length forecasts into online-calibrated *budget classes* parameterizing KV/compute constraints—an invariant interface for feasibility-checked admission and concurrency control, damping error amplification.
- **Hierarchical policy on budget-induced feasibility.** HERA optimizes over budget-induced feasibility sets: a meta-policy selects a resource envelope and weights; a constrained sub-policy composes admissible micro-batches/orderings—reducing search, enforcing feasibility, and remaining robust under bursty heavy tails.
- **Unified curriculum from prediction to control.** ORACLE uses refined bins with online calibration; HERA trains under staged arrivals (steady→periodic→bursty) toward steady-state control—tightening prediction–policy alignment, improving latency and deployability.

## 2 BACKGROUND

This section first reviews the architecture and inference pipeline of generative LLMs, then examines serving baselines and resource bottlenecks in online systems, and finally surveys scheduling strategies under dynamic traffic and workload uncertainty.

### 2.1 ARCHITECTURE & INFERENCE PIPELINE OF GENERATIVE LLMS

Generative LLMs predominantly adopt decoder-only Transformers—stacks of multi-head self-attention and feed-forward layers that generate tokens autoregressively Vaswani et al. (2017); Qiao et al. (2025); Dragunov et al. (2025). To mitigate decode-time memory bottlenecks, serving systems use paged KV-cache and continuous batching to raise VRAM utilization and throughput Kwon et al. (2023), and employ I/O-aware kernels such as FlashAttention to cut HBM↔SRAM traffic and avoid

materializing intermediates, thereby lowering attention bandwidth cost Dao et al. (2022). These systems optimizations surface prefill/decoding heterogeneity and expose low-level cost signals (compute, bandwidth, KV peaks) to upper-layer schedulers, enabling effective resource budgeting, batch construction, and policy learning.

## 2.2 SERVING PIPELINE, COST SURFACES, AND BOTTLENECKS IN LLM ONLINE SERVING

We view online LLM serving as a two-stage pipeline: *prefill*—a compute-heavy pass that encodes the prompt and writes per-layer KV in one shot—and *decode*—an autoregressive phase whose step time is constrained by VRAM capacity/layout and memory traffic Zhong et al. (2024); Agrawal et al. (2024); Bambhaniya et al. (2025); Liang et al. (2025); Zhang et al. (2025). On these cost surfaces, the execution/memory stack provides strong baselines: PagedAttention/vLLM and S-LoRA reduce fragmentation and enable adapter concurrency via paged KV pooling and continuous batching Kwon et al. (2023); Sheng et al. (2023), while Hugging Face TGI integrates dynamic batching, and tensor parallelism for general-purpose serving **?**. At the pipeline level, ORCA and successors decouple prefill and decoding across GPUs, add iteration-/token-level scheduling, and employ multi-level queues to mitigate head-of-line blocking Yu et al. (2022); Zhong et al. (2024). These advances span batch-level optimization, stage decoupling, and fine-grained preemption; yet production schedulers remain largely FCFS/watermark-driven, which under long-tailed or bursty arrivals inflates queueing delay and tail latency—underscoring that heuristic-only policies cannot jointly optimize latency and KV-cache efficiency Qiu et al. (2024).

## 2.3 SCHEDULING UNDER UNCERTAINTY

A central challenge in LLM serving is scheduling under uncertain output lengths and tightly coupled resources. In practice, systems such as vLLM and ORCA—and TGI for production inference—often default to FCFS for simplicity, which, under heavy tails and high load, induces head-of-line blocking and tail amplification Kwon et al. (2023); Yu et al. (2022). While classical SJF/SRPT is optimal with known sizes, it is brittle under misestimation and raises fairness concerns in multi-server settings. Recent work augments scheduling with prediction: Learning-to-Rank approximates SJF via relative length ordering Fu et al. (2024), LMaaS regresses generation length from input features to enable safer batching and adaptive sizing Cheng et al. (2024), and runtime predictors such as SSJF-Proxy and TRAIL estimate remaining length for preemptive control that balances reward and KV-cache cost Qiu et al. (2024); Shahout et al. (2024). Complementary directions—multi-bin batching Guldogan et al. (2024) and SLO-aware queue management Li et al. (2025)—improve constraint handling and mitigate FCFS degradation. Building on the need for interpretable, calibratable signals Guldogan et al. (2024), we propose ORACLE, which separates admission/budgeting from ordering/fairness calibration, yielding robust deployment gains under real traffic.

# 3 ATHENA-SERVE: AN INTELLIGENT SCHEDULING LLM SERVING SYSTEM VIA HORIZON-COST PREDICTION AND HIERARCHICAL RL

Guided by the above motivation, we present ATHENA-SERVE: a framework coupling horizon-aware budget signals with adaptive scheduling to optimize LLM inference under uncertain, bursty demand.

## 3.1 SYSTEM OVERVIEW

We formalize ATHENA-Serve as a predictive-control system comprising two operators. ORACLE defines a mapping $\mathcal{O} : \mathcal{X} \to \mathcal{B}$ from request features $\mathcal{X}$ to online-calibrated *budget constraints* $\mathcal{B}$ over KV-cache and compute resources. Within this induced constraint space, HERA implements a hierarchical policy that explicitly decouples budget enforcement from ordering and fairness calibration. This formulation integrates predictive estimation with hierarchical policy optimization, enabling provably stable, latency-sensitive scheduling under bursty, heavy-tailed workloads.

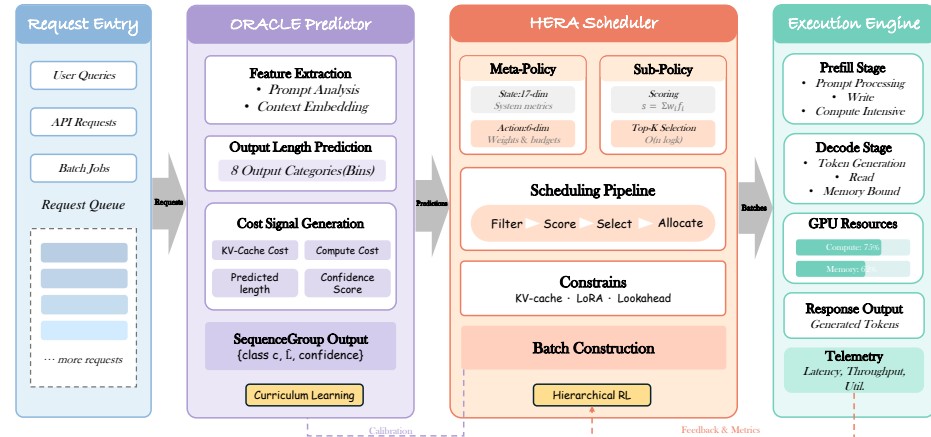

Figure 2: Overall framework of ATHENA-Serve.

## 3.2 ORACLE: OUTPUT-LENGTH-BASED REQUEST ASSESSMENT FOR COST-AWARE LLM EXECUTION

Recent work shows that efficient online LLM serving hinges on *request-level, forward-looking* cost awareness Fu et al. (2024); Cheng et al. (2024); Qiu et al. (2024). We introduce ORACLE, which predicts a per-request *horizon* and discretizes it into eight calibrated, interpretable budget classes, mapping to explicit KV-cache and compute *constraints* that endow the scheduler with direct semantics for feasibility, batching, and concurrency control.

### 3.2.1 PROBLEM FORMULATION AND COST MAPPING

This subsection formalizes ORACLE as a two-step map from request features to a calibrated horizon and then to KV/compute *budget* signals: $x \rightarrow \hat{L} \rightarrow \mathcal{B}(x)$, thus converting uncertain lengths into actionable semantics for feasibility checks, homogeneous batching, and concurrency control.

Given a request with prompt length $P$ (in tokens) and predicted output length $\hat{L}$, ORACLE returns

$$(\hat{c}, \hat{L}, \hat{p}) \in \{0, \dots, 7\} \times \mathbb{R}_+ \times [0, 1] \tag{1}$$

where $\hat{c}$ indexes a budget class (*instant→extreme*) and $\hat{p}$ is a calibration confidence used only to modulate small tolerance slack.

We consider a decoder-only transformer with $N$ layers, hidden size $H$, $A$ heads (head dim $d=H/A$), and element width $b$ (e.g., BF16/FP16: $b=2$ bytes). Under this model, we derive closed-form mappings from the predicted horizon $\hat{L}$ to KV-cache and compute *budgets*, which ground feasibility checks, homogeneous batching, and concurrency control.

**(i) KV-cache.** Per token and per layer, the KV-cache stores $K$ and $V$ vectors of total width $H$ across heads, yielding a *per-token* KV footprint

$$m_{\text{tok}} = 2\,N\,H\,b \quad \text{(bytes/token)} \tag{2}$$

For total sequence length $S=P+\hat{L}$, the *per-request* peak KV budget is

$$B_{\text{KV}}(P, \hat{L}) = S\,m_{\text{tok}} = 2\,N\,H\,b\,(P+\hat{L}) \tag{3}$$

Admission feasibility for active set $\mathcal{A}$ is the linear constraint

$$\sum_{i \in \mathcal{A}} B_{\text{KV}}(P_i, \hat{L}_i) + B_{\text{KV}}(P, \hat{L}) \leq M_{\text{KV,avail}} \tag{4}$$

**(ii) Compute (FLOPs).** Let $\alpha>0$ aggregate per-layer dense costs (projections/MLP) and $\beta>0$ aggregate per-layer attention costs; both are architecture constants. The *per-token* decode cost at context $s$ is

$$F_{\text{step}}(s) = N\big(\alpha H^2 + \beta\,s\,H\big) \tag{5}$$

which is strictly increasing in $s$. Hence the *prefill* and *decode* budgets are

$$F_{\text{pre}}(P) = \sum_{t=1}^{P} F_{\text{step}}(t) = N\Big(\alpha H^2 P + \beta H \frac{P(P+1)}{2}\Big) \tag{6}$$

$$F_{\text{dec}}(P, \hat{L}) = \sum_{t=1}^{\hat{L}} F_{\text{step}}(P + t - 1) = N\Big(\alpha H^2 \hat{L} + \beta H\Big(\hat{L}P + \frac{\hat{L}(\hat{L}-1)}{2}\Big)\Big) \tag{7}$$

Eqs. 6–7 formalize the familiar *quadratic* prefill and *linear+triangular* decode scaling.

**Lipschitz structure and tolerance.** From Eqs. 3–7, we have

$$\frac{\partial B_{\text{KV}}}{\partial \hat{L}} = 2NHb \tag{8}$$

$$\frac{\partial F_{\text{dec}}}{\partial \hat{L}} = N\Big(\alpha H^2 + \beta H\big(P + \hat{L} - \tfrac{1}{2}\big)\Big) \tag{9}$$

so both budgets vary *smoothly* with $\hat{L}$ and are locally Lipschitz. A small tolerance $\Delta$ in length prediction perturbs $(B_{\text{KV}}, F_{\text{dec}})$ only $O(\Delta)$, leaving batch composition and feasibility decisions invariant in a neighborhood. We therefore evaluate with a small token tolerance and treat boundary-neighbor classes as *budget-equivalent*.

ORACLE enables *tolerance-aware grouping* of cost-homogeneous requests. The confidence score $\hat{p}$ scales a minor slack on $\hat{L}$ (and consequently on all derived budgets) to bound the risk of under-provisioning without altering the budget class. Downstream, HERA enforces Eq. 4 while constructing near-homogeneous micro-batches and explicitly controlling the batchwise maximum context—thereby mitigating HoL and stabilizing tails.

### 3.2.2 MODEL ARCHITECTURE AND TRAINING SETUP

**Architecture.** ORACLE employs a "feature extractor + lightweight heads" design, using a distilled LLaMA-based model(325M) as the semantic extractor. Feature representations are processed through a Global-Context (GC) block and an MLP bottleneck. The model outputs an 8-class head and a regression head, with stability ensured via EMA during training.

**Curriculum and Sampling.** ORACLE is trained with a five-stage, coarse-to-fine curriculum with difficulty-gated sampling. LoRA regularization, EMA smoothing, and dynamic loss weights stabilize convergence under finer bins. A formal specification of the curriculum $(\{\phi_s\}, \{q_s\}, \{\mathcal{L}_s\})$ appears in Appendix A.1.

**Losses and Regularization.** A composite objective (CE, focal, consistency/adjacency regularizers, curriculum term, and long-tail reweighting) balances separability, ordinality, and robustness.

## 3.3 HERA: HIERARCHICAL RL FOR HORIZON-AWARE EXECUTION RESOURCE ALLOCATION

### 3.3.1 DESIGN GOALS AND TRIPLE AWARENESS: HORIZON, SYSTEM, AND GLOBAL LOAD

At decision epoch $t$, with system state $s_t$ (GPU/VRAM utilizations, queue statistics, burstiness features, etc.) and ORACLE cues $\{r_i\}$ for waiting requests (horizon–to–budget signals), HERA chooses a *per-tick resource envelope* $b_t$ and a *weight vector* $\theta_t = (\alpha, \beta, \gamma)$, and induces a scheduling policy $\pi_t$ by solving

$$(b_t^\star, \theta_t^\star, \pi_t^\star) \in \underset{b_t \in \mathcal{B}(s_t),\, \theta_t \in \Theta,\, \pi \in \mathcal{F}(b_t, \{r_i\})}{\arg\min} \mathbb{E}\big[\ell_{\text{lat}}(\pi) \mid s_t\big] \tag{10}$$

where $\mathcal{F}(b_t, \{r_i\}$ enforces feasibility under the envelope $b_t$ and the request-level budget cues $r_i$.

Its value kernel comprises three complementary awarenesses: *horizon-aware*—calibrated class medians enable budgeted admission, deliberately postponing long-budget jobs while prioritizing short ones to accelerate KV release; *system-aware*—admission intensity, batch composition,

and decoding concurrency are co-regulated within hard resource envelopes; and *global-load-aware*—admission/packing is decoupled from queue ordering and fairness calibration, with enforced outcomes (later for budget shortfalls, ignore for anomalies) to preclude congestion and starvation. Operating on interpretable hierarchical scores over discretized budgets, HERA dampens horizon-prediction noise and remains stable under bursty, heavy-tailed demand.

---

**Algorithm 1** HERA Scheduling per Tick

---

**Require:** Waiting, running, and swapped queues; scheduler configuration; ORACLE feature cache

1: $W \leftarrow$ snapshot of waiting queue; remove invalid groups
2: $C \leftarrow$ apply lightweight prefilter for LoRA/illegal groups
3: For each group $g \in C$, compute the scheduling score:
4: $\sigma_i = \alpha \, \text{priority}_i + \beta \, w_i + \gamma \, h_i$, where $w_i$ is waiting time and $h_i$ is the horizon cue from ORACLE.
5: $T \leftarrow$ select top-K groups by score
6: For each group in $T$, verify feasibility against available resource budgets (KV-cache, compute, memory); check LoRA concurrency and latency constraints
7: If feasible, allocate resources and update corresponding budgets; otherwise, mark for deferral or ignore based on resource availability
8: Rebuild waiting queue from scheduled and ignored groups
9: Collect throughput and TTFT metrics; adjust sub-policy weights to balance latency, priority, and resource utilization

**Ensure:** Prefill, running, and swapped decisions for this tick

---

### 3.3.2 STATE, ACTIONS, AND HIERARCHICAL DECOUPLING

**State.** At each decision epoch, the controller consumes a unified summary comprising (i) system-level telemetry (VRAM/KV pressure, compute/bandwidth utilization), (ii) workload descriptors for waiting and newly arrived requests (queue depth/growth, horizon–budget, waiting-time profile, burstiness indicators), and (iii) service-quality signals (TTFT/E2E aggregates with EMA baselines).

**Actions and executors.** We decompose the action $a_t$ into two groups, emitted by a single actor and routed to different executors:

**Meta-policy (resources and admission).** Maps $s_t$ to $(b_t, \theta_t)$: it adjusts the effective resource envelope $b_t$ within strict caps—expanding under computational slack, contracting under memory or latency pressure—and emits $\theta_t$ to shape the hierarchical score, balancing priority, waiting, and horizon preference so that feasibility, batching, and concurrency remain aligned with the evolving state.

**Sub-policy (ordering and realization).** Conditioned on $(b_t, \theta_t)$ and ORACLE cues, it admits a feasible set of requests and orders them by the hierarchical score, marking infeasible items for deferral or ignore, thereby realizing per-tick scheduling under the budgeted envelope.

**Decoupling benefits.** (i) Separates global feasibility/admission from local ordering/trade-offs, reducing one-step credit-assignment variance; (ii) clamps all actions within physical limits for safe gates and rollbacks; (iii) cold-starts with a near-FCFS safety bias and transitions stably to superior policies via gated online data.

### 3.3.3 REWARD SHAPING WITH LATENCY GATING

To avoid destabilization from noisy windows and missing data, HERA uses log-ratio terms with $\tanh$ stabilization, asymmetric EMA baselines, and source-aware gating: latency signals open only when TTFT originates from the client bridge and throughput exceeds a threshold. Let windowed throughput/latency (p95) be $T_t/L_t$ and EMA baselines $\bar{T}/\bar{L}$. Define

$$r_{\text{thr}} = \tanh\big(\log(T_t/\bar{T})\big), \; r_{\text{lat}} = -\tanh\big(\log(L_t/\bar{L})\big) \tag{11}$$

We add the completion rate and the queue p95 terms to form

$$R_t = r_{\text{thr}} + g_{\text{lat}} \cdot r_{\text{lat}} + r_{\text{rps}} + r_{\text{q}} \tag{12}$$

where $r_{\text{thr}}$ denotes throughput reward, $g_{\text{lat}} \cdot r_{\text{lat}}$ denotes a gate-modulated p95-TTFT penalty (linearly increasing with reliable client-bridge samples and attenuated under fallback), $r_{\text{rps}}$ denotes the request-completion reward, and $r_{\text{q}}$ denotes the queue-wait p95 penalty.

Baselines utilize asymmetric EMA, and update only when the TTFT source is the client bridge. The gate coefficient increases linearly with consecutive high-quality bridge frames.

### 3.3.4 CURRICULUM-GUIDED ONLINE TRAINING AND SAFE ROLLOUT

**Curriculum-guided learning.** We train under a staged arrival curriculum—that progressively increases non-stationarity and load. This progression lets the policy internalize stable admission/ordering in controlled regimes, then adapt to bursty, near-OOM conditions, yielding deployable behavior. Formal definitions and parameterizations appear in Appendix **??**.

**Closed loop with ORACLE.** HERA consumes ORACLE's discretized horizon signals, refreshed online via EMA, as calibrated cost proxies. This loop decouples resource budgeting from request ordering, improving robustness to prediction error and distributional shift without incurring expensive global re-ranking.

## 4 EVALUATION

In this section, we present evaluation results to isolate the effect of *horizon-aware* prediction and hierarchical scheduling on latency under standardized workloads and identical hardware/software settings.

**Setup and Implementation.** Our experiments use vLLM 0.8.0 on a single server (NVIDIA A40 48 GB, Intel Xeon Gold 6348 24C/48T, 377 GiB RAM, CUDA 12.4). Inference is exposed via the vLLM OpenAI API with Llama-7B-Chat on one GPU. The workload comprises 10k first-turn ShareGPT prompts, disjoint from those used to train ATHENA-Serve. We replay identical request traces across systems under controlled arrival regimes with varying intensity and burstiness.

**Baselines.** We compare four serving schedulers on the same model and hardware: (i) vLLMKwon et al. (2023) with its default FCFS policy; (ii) vLLM+Oracle, an SJF-like variant (ablation) that uses our horizon predictor without hierarchical control; (iii) TGIHug (2025) with its default scheduler; and (iv) ATHENA-Serve (ours), which couples horizon-aware admission with batch construction and feasibility checks. This design controls for model and system factors so that differences can be attributed to scheduling structure.

### 4.1 PREDICTION ACCURACY AND SCHEDULING SEMANTICS

We first evaluate the predictive fidelity of ORACLE and make explicit the *scheduling semantics* it enables—i.e., how predictions translate into budgeted admission and batch construction under fixed hardware and decoding settings.[1]

We discretize the continuous, aleatorically uncertain output length $L$ (decode tokens) into $K{=}8$ *horizon classes* $\mathcal{H}{=}\{instant, \ldots, extreme\}$ ordered by increasing resource budget. This aligns evaluation with how serving decisions are made in practice—by *budgets*, not exact lengths.

Viewed through a scheduling lens, Fig. 3a's reliability curve (accuracy vs. confidence, with per-bin support) links confidence to operational risk: well-calibrated regions permit tight token budgets, whereas miscalibrated or data-sparse bins warrant proportional safety slack to avoid under-provisioning. Fig. 3b complements this with per-class precision/recall/F1 that vary smoothly across the eight length bins, with errors dominated by *adjacent-bin confusions* rather than long-range mistakes. Since prefill/decoding time and KV-cache peaks are locally Lipschitz in output length $L$, we evaluate the oracle under $\pm 20$ token tolerance and treat boundary-neighbor predictions as budget-feasible. The actionable unit is thus a *tolerance-aware budget cluster*, not a single bin: confidence

---

[1]Cross-model comparisons are not meaningful here: output length distributions and token-to-time mappings depend on the specific LLM, decoding parameters, and serving stack. We therefore restrict prediction analysis to the same model/runtime as our scheduler.

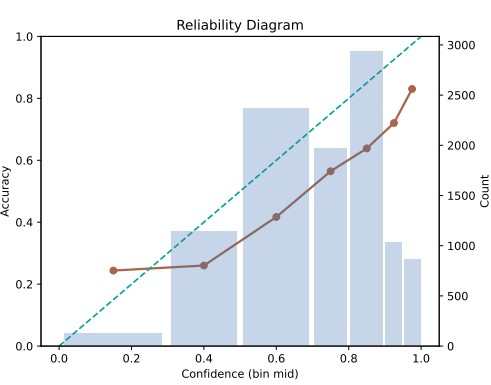 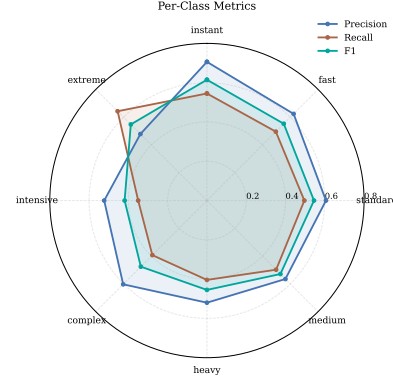

(a) Reliability curve: accuracy vs. confidence.

(b) Calibration of Oracle performance.

Figure 3: Reliability and class-wise structure of Oracle horizon predictions.

becomes a knob for per-request slack, batch composition and parallelism remain stable under small length perturbations.

## 4.2 MITIGATING HEAD-OF-LINE UNDER HEAVY-TAILED HORIZONS

Head-of-line (HoL) in decode-stepped LLM serving arises because each step's duration is pinned by the *largest* active horizon in a batch; mixing short and long requests then amplifies queueing for the majority. To study Athena's impact, we examine the empirical mixture of output-length categories (Fig. 4a) alongside the CDF of TTFT under four schedulers (Fig. 4b).

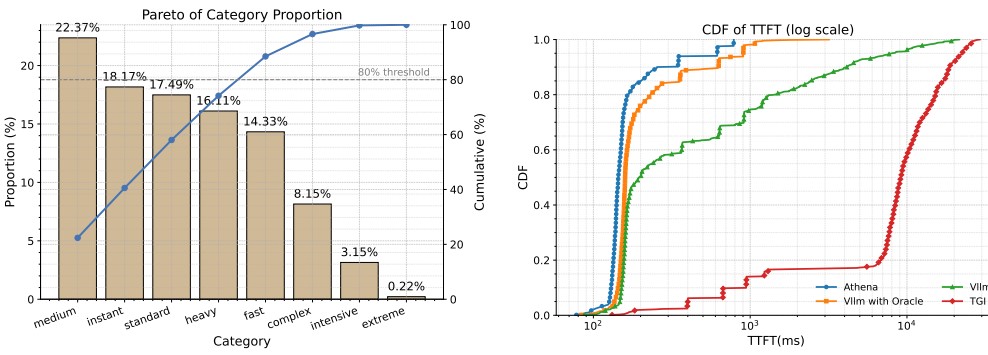

(a) Pareto distribution of output-length categories (long-tail effect).

(b) CDF of Time-to-First-Token (TTFT) under different schedulers.

Figure 4: HoL mitigation analysis: heavy-tailed output lengths (left) create systematic head-of-line delays, while Athena's horizon-aware scheduling reduces TTFT across the distribution (right).

Fig. 4a exhibits a Pareto-like output–length mix: a few mid-length bins carry most mass, yet a thin heavy tail (complex/intensive/extreme) persists. This is structurally consequential: FIFO schedule repeatedly co-locates long-horizon jobs with short ones, so the batch's $\ell_\infty$ horizon pins each decode step and induces systematic HoL. The objective is therefore not to "favor short jobs", but to *minimize the maximum effective horizon per active batch* under memory/parallelism constraints, keeping step times near-uniform. ATHENA operationalizes this by converting length predictions (with tolerance-aware budgets) into admission ceilings and near-homogeneous micro-batches, turning the heavy-tailed mix into a controllable budget envelope. Consistent with this mechanism, the TTFT CDF in Fig. 4b is left-shifted relative to vLLM/TGI across most of the support, with the largest gains occurring where traffic is concentrated. vLLM+Oracle narrows the median through size awareness, but without concurrency control, long requests still introduce stragglers into short-serving batches.

### 4.3 Evaluating hroughput–Latency Trade-off Under Varied Workloads

We evaluate the throughput–latency behavior under several fixed QPS levels. On the server side, we use the HERA-enabled version of vLLM with the lightweight intelligent scheduler and SAC turned on. On the client side, we use the 'hera light' entrypoint to generate requests according to a Poisson process at fixed rates (QPS = 50/100/150/200). All requests are drawn from the same served Alpaca dataset, and we keep sampling parameters (temperature, top-p, max tokens, etc.) fixed across experiments.

During evaluation, the scheduler aggregates metrics over fixed time windows (e.g., 2 seconds), computing the throughput within each window (scheduled tokens/s) and the first-token P95 latency (p95 latency). We apply a moving average to both sequences to obtain smoothed curves of throughput and P95 latency as a function of step, and we plot on the latency axis a reference line corresponding to the SLO = 0.425 s, derived as 5× TTFT measured from single-request runs.

The four subfigures jointly characterize how, under different load intensities, the HERA intelligent scheduler trades off high throughput against adherence to the latency SLO. At low QPS (qps50, qps100), the P95 latency is clearly below the SLO, and throughput scales approximately linearly with load. As QPS increases to 150 and 200, throughput gains gradually saturate, while P95 latency shifts upward overall and more frequently approaches or exceeds the SLO window-by-window, illustrating the throughput–latency trade-off as the system operates near its capacity limit.

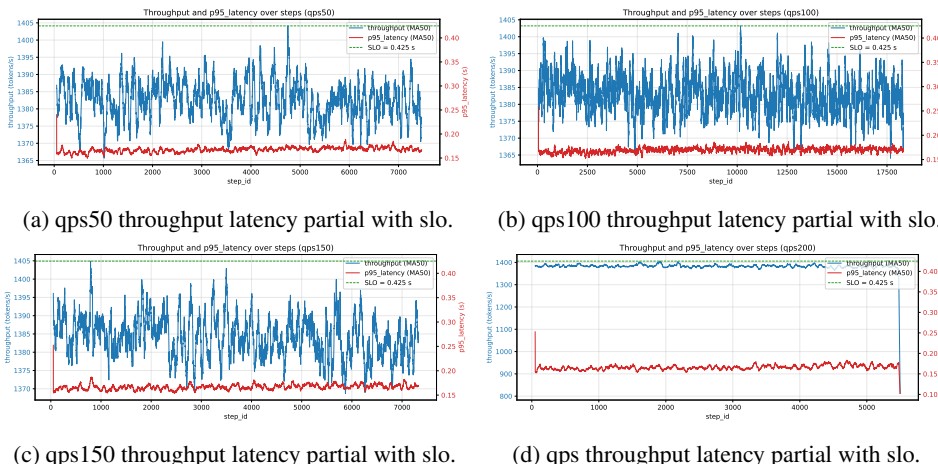

(a) qps50 throughput latency partial with slo.   (b) qps100 throughput latency partial with slo.

(c) qps150 throughput latency partial with slo.   (d) qps throughput latency partial with slo.

Figure 5: Throughput and P95 latency of the HERA intelligent scheduler under different load levels. The blue curve shows the moving average of throughput (tokens/s) within each RL reward window, and the red curve shows the corresponding moving average of the first-token P95 latency (p95latency) over the same window. The green dashed line denotes the SLO (0.425 s), set to 5× the TTFT measured from a single-request benchmark.

## 5 Discussion and Future Work

Deployable LLM serving rests on translating uncertain horizons into *principled* resource semantics. Oracle calibrates horizons into discretized KV/compute *budget constraints*, enabling feasibility-checked admission, homogeneous batching, and interpretable concurrency control. Conditioning on these constraints, Hera optimizes a *hierarchical policy* that decouples admission/budget enforcement from ordering/fairness and co-tunes admission intensity with decoding concurrency, safeguarding tail latency under heavy-tailed, and bursty arrivals. The resulting budgeted triad—horizon, system, global load—accounts for the observed robustness.

Beyond empirical gains, the framework offers two conceptual takeaways: (i) *Budget semantics* provide an invariant contract between prediction and scheduling, avoiding the noise amplification of length- or rank–only policies; (ii) *hierarchical decoupling* matters: separating global feasibility/budgeting from local ordering/fairness yields tractable search, lower-variance credit assignment, feasibility by construction, and stable on-policy improvement—properties that travel across run-

times and workloads. Extending to multi-node/heterogeneous clusters and token-level preemption, and integrating uncertainty sets with violation/regret guarantees, are promising directions.

**Reproducibility Statement.** We facilitate reproducibility by specifying the modeling assumptions, objective, workloads, and evaluation protocol in the main paper and appendix. The *Problem Formulation and Cost Mapping* subsection gives closed-form budgets for KV-cache and compute and states all architectural constants and tolerance rules; *HERA: Hierarchical RL for Horizon-aware Execution Resource Allocation* formalizes the per-epoch optimization and provides an executable pseudocode abstraction (Algorithm 1), together with the *Reward Shaping with Latency Gating* equations and gating conditions. Workloads are generated by a seeded NHPP with five arrival regimes—foundation, fluctuation, burst, complex, stress—whose formal definitions and parameterizations appear in Appendix **??**. The *Formal Curriculum for* ORACLE in Appendix A.1 defines the coarse→fine label maps, difficulty-gated sampling, loss composition, and EMA smoothing. We report metric definitions (TTFT, E2E, percentiles), measurement windows, and baseline settings under a shared hardware/runtime configuration.

**Ethics Statement.** This work does not involve human subjects, sensitive personal data, or private information, and all experiments are conducted on publicly available LLM models and synthetic arrival traces. The methodology focuses on system-level scheduling and resource allocation, without modifying model content, generating potentially harmful insights, or addressing demographic attributes. No proprietary datasets are released or required; workload regimes are mathematically defined and reproducible as detailed in the appendix. The study complies with standards of research integrity, legal use of computational resources, and responsible reporting of limitations (e.g., calibration dependence, cluster extensions). We thus do not foresee risks regarding fairness, bias, or misuse, but emphasize that the contributions should be interpreted strictly within the context of system performance and efficiency in LLM serving.

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

## LLMs Usage Disclosure

The authors disclose that Large Language Models (LLMs) were used only to aid or polish writing.

## A  Appendix

### A.1  Formal Curriculum for Oracle

**Setup.**  Given features $x$ (prompt/length metadata) and true output length $L \in \mathbb{R}_+$, Oracle learns $f_\theta : x \mapsto \hat{y}$ with $\hat{y} \in \{0, \dots, K-1\}$ (final $K = 8$ budget classes). Training proceeds in stages $s = 0, \dots, S-1$, each specifying (i) a label map $\phi_s : \mathbb{R}_+ \to \{0, \dots, K_s-1\}$ (coarse→fine) and (ii) a sampling law $q_s$ over the dataset. Stages instantiate a coarse-to-fine sequence with stage-specific focus and sampling, with EMA and early stopping enabled as configured.

**Difficulty-gated sampling.**  A difficulty score $d(x) \in [0, 1]$ aggregates boundary proximity, length ambiguity, semantic complexity, and class rarity; curriculum thresholds $\delta_s \in \{0.3, 0.6, 0.8, 1.0\}$ unveil easy→all subsets with $q_s$ restricted to $\{x : d(x) \leq \delta_s\}$. Boundary proximity is computed against fixed bin edges; rarity weights emphasize scarce classes.

**Structure-preserving, adaptive reweighting.**  Within stage $s$, instance weights blend structure-preserving intra-class shape, inter-class balance, and performance-adaptive terms:

$$w^{\text{final}} \propto \underbrace{\rho \, w^{\text{intra}}}_{\text{structure}} + \underbrace{(1-\rho) \cdot \left(0.6 \, w^{\text{inter}} + 0.4 \, w^{\text{adapt}}\right)}_{\text{balance + adapt}} \tag{13}$$

with $w^{\text{adapt}}$ updated from per-class metrics (recall/F1) to upweight hard or confused classes; stage multipliers emphasize later (finer) phases.

**Stage objective and smoothing.**  Each stage optimizes the expected stage-appropriate loss

$$\min_\theta \ \mathbb{E}_{x \sim q_s} \left[ \mathcal{L}_s \big( f_\theta(x), \, \phi_s(L) \big) \right] \tag{14}$$

with *classification*-only in the final refinement stage and optional EMA smoothing for stability under finer labels. :contentReference[oaicite:10]index=10 :contentReference[oaicite:11]index=11

**Summary.**  The curriculum is defined by the triplet $\big(\{\phi_s\}, \{q_s\}, \{\mathcal{L}_s\}\big)$: coarse→fine label maps, difficulty-gated sampling, and structure-preserving adaptive reweighting—augmented by EMA—yielding calibrated eight-class budgets aligned with downstream scheduling.

### A.2  Regret and stability analysis of hierarchical budget-constrained RL

We provide three formal results showing that:

1. after discretizing the meta-action (resource envelope + weights), the meta-policy admits a standard online no-regret guarantee;

2. when the reward is Lipschitz in the action, the approximation error from discretizing the continuous action space into a finite grid is controlled;

3. under budget-constrained admission and concurrency control, the resulting queueing process is positive recurrent / stable.

### A.2.1  Setup and assumptions

At each scheduling decision epoch $t = 1, 2, \dots, T$, the system is in state $s_t \in \mathcal{S}$, and the meta-policy chooses a meta-action $a_t = (b_t, \theta_t) \in \mathcal{A} \subset \mathbb{R}^d$, where $b_t$ denotes a resource envelope (admission budget, concurrency budget, etc.), and $\theta_t = (\alpha_t, \beta_t, \gamma_t)$ are the weights used in the hierarchical scoring rule.

Given $(s_t, a_t)$ and the request-level budget information from ORACLE, the HERA sub-policy filters and orders the feasible set under the resource envelope $b_t$. The system returns a scalar reward $R_t(a_t) \in \mathbb{R}$, which is a combination of throughput, latency, and queueing terms (matching $r_{\text{thr}}, r_{\text{lat}}, r_{\text{rps}}, r_q$ in the main paper).

We now state several mild assumptions that are compatible with the construction in the main text.

**Assumption 1 (Bounded reward).** There exists a constant $R_{\max} > 0$ such that for all $t$ and all $a \in \mathcal{A}$ the one-step reward satisfies

$$|R_t(a)| \leq R_{\max}. \tag{15}$$

Equivalently, the rescaled reward

$$r_t(a) = \frac{R_t(a) + R_{\max}}{2R_{\max}} \tag{16}$$

satisfies $r_t(a) \in [0, 1]$.

In the main paper, the reward is built from several $\tanh(\log(\cdot))$ terms and bounded queue / throughput terms. Each component is supported on a compact interval, so the overall reward is automatically bounded.

**Assumption 2 (Lipschitz continuity in the meta-action).** There exists a constant $L > 0$ such that for all $t$ and all $a, a' \in \mathcal{A}$,

$$\left| r_t(a) - r_t(a') \right| \leq L \, \|a - a'\|_2. \tag{17}$$

ORACLE maps the predicted length $\hat{L}$ into KV and compute budgets through smooth, Lipschitz mappings. The reward is a smooth function (linear combinations, $\tanh$, etc.) of these budgeted quantities such as length, latency, and throughput. Hence the overall mapping from action to reward is Lipschitz.

**Definition (Discretization of the meta-action space).** Let $\mathcal{A} \subset \mathbb{R}^d$ be compact. Given a resolution $\varepsilon > 0$, a finite set $\widetilde{\mathcal{A}} \subset \mathcal{A}$ is called an $\varepsilon$-grid of $\mathcal{A}$ if for any $a \in \mathcal{A}$ there exists a projection point $\Pi(a) \in \widetilde{\mathcal{A}}$ such that

$$\|a - \Pi(a)\|_2 \leq \varepsilon. \tag{18}$$

In deployment, actions of the meta-policy lie in a bounded interval and are quantized into finitely many discrete levels, which matches this definition exactly.

A.2.2 NO-REGRET LEARNING WITH DISCRETE META-ACTIONS

After discretizing the meta-action space, we can view the problem as online learning over the finite set $\widetilde{\mathcal{A}}$. The following proposition shows that running the standard exponential-weights (Hedge) algorithm yields $O(\sqrt{T})$ regret.

**Proposition 1 (No-regret learning over discrete meta-actions).** Suppose Assumption 1 holds, and let $\widetilde{\mathcal{A}}$ be a finite set of discrete meta-actions with $K = |\widetilde{\mathcal{A}}|$. Consider any sequence of states $\{s_t\}_{t=1}^T$ and any sequence of reward functions $\{r_t(\cdot)\}_{t=1}^T$ with $r_t : \widetilde{\mathcal{A}} \to [0, 1]$ arbitrary.

Let the meta-policy run the Hedge algorithm over $\widetilde{\mathcal{A}}$:

- Initialization: for all $a \in \widetilde{\mathcal{A}}$, set $w_1(a) = 1$.
- For $t = 1, \dots, T$:
  1. Construct the action distribution

  $$p_t(a) = \frac{w_t(a)}{\sum_{a' \in \widetilde{\mathcal{A}}} w_t(a')}. \tag{19}$$

  2. Sample $a_t \sim p_t(\cdot)$ and execute this action.

3. Observe the rewards $r_t(a)$ for all actions (equivalently, the losses $\ell_t(a) = 1 - r_t(a)$).

4. Update the weights:

$$w_{t+1}(a) = w_t(a) \exp(-\eta\,\ell_t(a)), \tag{20}$$

where $\eta > 0$ is the learning rate.

Then for any fixed comparator action $a^\star \in \widetilde{\mathcal{A}}$, if we choose

$$\eta = \sqrt{\frac{2\log K}{T}}, \tag{21}$$

the expected regret satisfies

$$\mathbb{E}\Big[\sum_{t=1}^{T}\big(r_t(a^\star) - r_t(a_t)\big)\Big] \leq \sqrt{2T\log K}. \tag{22}$$

In particular, the average per-round regret converges to zero as $T \to \infty$,

$$\frac{1}{T}\,\mathrm{Regret}(T) \to 0. \tag{23}$$

**Proof of Proposition 1.** For convenience, we work with losses $\ell_t(a) = 1 - r_t(a) \in [0,1]$ instead of rewards. The randomized loss incurred in round $t$ is $\ell_t(a_t)$, and its conditional expectation under $p_t$ is

$$L_t = \sum_{a\in\widetilde{\mathcal{A}}} p_t(a)\,\ell_t(a) = \mathbb{E}[\ell_t(a_t) \mid p_t]. \tag{24}$$

The Hedge update is

$$w_{t+1}(a) = w_t(a) \exp(-\eta\,\ell_t(a)). \tag{25}$$

Let $W_t = \sum_a w_t(a)$ denote the sum of weights. We will upper- and lower-bound $\log W_{T+1}$.

**(1) Upper bound on $\log W_{T+1}$.** From the update,

$$W_{t+1} = \sum_a w_t(a) \exp(-\eta\,\ell_t(a)). \tag{26}$$

Using $e^{-x} \leq 1 - x + \frac{x^2}{2}$ for $x \in [0,1]$,

$$W_{t+1} \leq \sum_a w_t(a)\Big(1 - \eta\,\ell_t(a) + \frac{\eta^2}{2}\,\ell_t(a)^2\Big) \tag{27}$$

$$\leq W_t\Big(1 - \eta\sum_a p_t(a)\,\ell_t(a) + \frac{\eta^2}{2}\sum_a p_t(a)\,\ell_t(a)^2\Big) \tag{28}$$

$$\leq W_t\Big(1 - \eta L_t + \frac{\eta^2}{2}\Big), \tag{29}$$

where we used $\ell_t(a)^2 \leq \ell_t(a) \leq 1$. Taking logs and using $\log(1+x) \leq x$ for $x > -1$,

$$\log W_{t+1} - \log W_t \leq -\eta L_t + \frac{\eta^2}{2}. \tag{30}$$

Summing from $t = 1$ to $T$,

$$\log W_{T+1} - \log W_1 \leq -\eta\sum_{t=1}^{T} L_t + \frac{\eta^2 T}{2}. \tag{31}$$

Since $w_1(a) = 1$ for all $a$, we have $W_1 = K$ and $\log W_1 = \log K$.

**(2) Lower bound on $\log W_{T+1}$.** For any fixed comparator $a^\star \in \widetilde{\mathcal{A}}$,

$$w_{T+1}(a^\star) = w_1(a^\star) \exp\Big(-\eta\sum_{t=1}^{T}\ell_t(a^\star)\Big) = \exp\Big(-\eta\sum_{t=1}^{T}\ell_t(a^\star)\Big), \tag{32}$$

since $w_1(a^\star) = 1$. As $W_{T+1} \geq w_{T+1}(a^\star)$, we obtain

$$\log W_{T+1} \geq -\eta\sum_{t=1}^{T}\ell_t(a^\star). \tag{33}$$

**(3) Combining the bounds.** Combining the upper and lower bounds,

$$-\eta \sum_{t=1}^{T} \ell_t(a^\star) \ \leq \ \log W_{T+1} \ \leq \ \log K - \eta \sum_{t=1}^{T} L_t + \frac{\eta^2 T}{2}. \tag{34}$$

Rearranging,

$$\sum_{t=1}^{T} L_t - \sum_{t=1}^{T} \ell_t(a^\star) \ \leq \ \frac{\log K}{\eta} + \frac{\eta T}{2}. \tag{35}$$

Taking expectations and using $\mathbb{E}[\ell_t(a_t)] = L_t$,

$$\mathbb{E}\Big[ \sum_{t=1}^{T} \ell_t(a_t) \Big] - \sum_{t=1}^{T} \ell_t(a^\star) \ \leq \ \frac{\log K}{\eta} + \frac{\eta T}{2}. \tag{36}$$

Since $\ell_t(a) = 1 - r_t(a)$, we have

$$\sum_{t=1}^{T} \ell_t(a_t) - \sum_{t=1}^{T} \ell_t(a^\star) = \sum_{t=1}^{T}(1 - r_t(a_t)) - \sum_{t=1}^{T}(1 - r_t(a^\star)) \tag{37}$$

$$= \sum_{t=1}^{T} \big( r_t(a^\star) - r_t(a_t) \big). \tag{38}$$

Hence

$$\mathbb{E}\Big[ \sum_{t=1}^{T}(r_t(a^\star) - r_t(a_t)) \Big] \ \leq \ \frac{\log K}{\eta} + \frac{\eta T}{2}. \tag{39}$$

Choosing $\eta = \sqrt{2 \log K / T}$ yields

$$\mathbb{E}\Big[ \sum_{t=1}^{T}(r_t(a^\star) - r_t(a_t)) \Big] \ \leq \ \sqrt{2T \log K}, \tag{40}$$

which proves the proposition. $\qquad\square$

### A.2.3 APPROXIMATION ERROR BETWEEN CONTINUOUS ACTIONS AND A DISCRETE GRID

We now show that when the reward is Lipschitz in the action, replacing the continuous action space $\mathcal{A}$ with an $\varepsilon$-grid $\widetilde{\mathcal{A}}$ incurs at most $L\varepsilon T$ total reward error.

**Proposition 2 (Approximation error from continuous to discrete actions).** Suppose Assumption 2 holds and that $\widetilde{\mathcal{A}}$ is an $\varepsilon$-grid of $\mathcal{A}$. For each $a \in \mathcal{A}$ let $\Pi(a) \in \widetilde{\mathcal{A}}$ be the closest grid point, so that

$$\|a - \Pi(a)\|_2 \leq \varepsilon. \tag{41}$$

Define the continuously optimal meta-action

$$a^\star_{\mathrm{cont}} \in \arg\max_{a \in \mathcal{A}} \sum_{t=1}^{T} r_t(a), \tag{42}$$

and the discretely optimal meta-action

$$a^\star_{\mathrm{disc}} \in \arg\max_{a \in \widetilde{\mathcal{A}}} \sum_{t=1}^{T} r_t(a). \tag{43}$$

Then

$$\sum_{t=1}^{T} r_t(a^\star_{\mathrm{cont}}) \ \leq \ \sum_{t=1}^{T} r_t(a^\star_{\mathrm{disc}}) + L\varepsilon T. \tag{44}$$

**Proof of Proposition 2.**  By the definition of $a^\star_{\text{disc}}$, for any $a \in \mathcal{A}$,

$$\sum_{t=1}^{T} r_t(a^\star_{\text{disc}}) \geq \sum_{t=1}^{T} r_t(\Pi(a)). \tag{45}$$

In particular, taking $a = a^\star_{\text{cont}}$,

$$\sum_{t=1}^{T} r_t(a^\star_{\text{disc}}) \geq \sum_{t=1}^{T} r_t\big(\Pi(a^\star_{\text{cont}})\big). \tag{46}$$

On the other hand, by Lipschitz continuity, for each $t$,

$$\big|r_t(a^\star_{\text{cont}}) - r_t(\Pi(a^\star_{\text{cont}}))\big| \leq L\,\|a^\star_{\text{cont}} - \Pi(a^\star_{\text{cont}})\|_2 \leq L\varepsilon. \tag{47}$$

Thus

$$r_t(a^\star_{\text{cont}}) \leq r_t(\Pi(a^\star_{\text{cont}})) + L\varepsilon. \tag{48}$$

Summing over $t = 1, \ldots, T$,

$$\sum_{t=1}^{T} r_t(a^\star_{\text{cont}}) \leq \sum_{t=1}^{T} r_t\big(\Pi(a^\star_{\text{cont}})\big) + L\varepsilon T. \tag{49}$$

Combining with the inequality above yields

$$\sum_{t=1}^{T} r_t(a^\star_{\text{cont}}) \leq \sum_{t=1}^{T} r_t(a^\star_{\text{disc}}) + L\varepsilon T, \tag{50}$$

which proves the proposition. $\square$

**Corollary (Regret relative to the continuous optimum).**  Under the conditions of Propositions 1 and 2, let the meta-policy run Hedge on the $\varepsilon$-grid $\widetilde{\mathcal{A}}$ with $K = |\widetilde{\mathcal{A}}|$. Then for any $T$,

$$\mathbb{E}\Big[ \sum_{t=1}^{T} \big(r_t(a^\star_{\text{cont}}) - r_t(a_t)\big) \Big] \leq \sqrt{2T \log K} + L\varepsilon T. \tag{51}$$

The total difference is decomposed into:

- the gap between the continuous and discrete optima, bounded by $L\varepsilon T$ by Proposition 2;
- the gap between the discrete optimum and the algorithm, bounded by $\sqrt{2T \log K}$ by Proposition 1.

### A.2.4  QUEUE STABILITY UNDER BUDGET-CONSTRAINED CONTROL

We now introduce a standard queueing model and show that with budget-constrained admission and concurrency control, as long as the load is below capacity, the backlog process is stable.

**Queueing model.**  We model scheduler ticks as a discrete-time process $t = 1, 2, \ldots$. Let

- $Q_t$: total backlog at time $t$ (e.g., number of waiting / in-service requests, or total remaining decode tokens);
- $A_t \geq 0$: total work arriving in the interval $[t, t+1)$;
- $S_t \geq 0$: total work processed in $[t, t+1)$, determined by the meta-action $a_t$ and the corresponding sub-policy.

The queue evolves according to

$$Q_{t+1} = [Q_t - S_t]_+ + A_t, \qquad [x]_+ = \max\{x, 0\}. \tag{52}$$

We make the following standard assumptions on arrivals and service.

**Assumption 3 (Arrival process).** The arrival process $\{A_t\}$ is independent of the scheduling decisions and is i.i.d. with

$$\mathbb{E}[A_t] = \lambda, \qquad \mathbb{E}[A_t^2] < \infty. \tag{53}$$

**Assumption 4 (Service upper bound and "non-idling" resource envelope).** There exists a constant $S_{\max} > 0$ such that for all $t$ almost surely,

$$0 \leq S_t \leq S_{\max}. \tag{54}$$

Moreover, there exist constants $\mu > 0$ and $Q_0 \geq 0$ such that for all $t$, if $Q_t = q \geq Q_0$, then

$$\mathbb{E}[S_t \mid Q_t = q] \ \geq \ \mu. \tag{55}$$

Intuitively, the resource envelope $b_t$ both respects hardware capacity (yielding $S_{\max}$) and avoids idling when the queue is large; in the large-backlog region the expected service rate is lower bounded by $\mu$.

**Assumption 5 (Subcritical load).** The long-term average arrival rate satisfies

$$\lambda < \mu. \tag{56}$$

This is the classical "arrival rate < service capacity" stability condition, expressed here in budget units.

**Proposition 3 (Positive recurrence / stability of the backlog).** Under Assumptions 3–5, the Markov chain $\{Q_t\}$ defined by the queueing recursion above is positive recurrent. In particular, there exists a constant $C < \infty$ such that in steady state

$$\mathbb{E}[Q] \leq C. \tag{57}$$

**Proof of Proposition 3 (Lyapunov drift).** We apply the Foster–Lyapunov drift criterion. Define the Lyapunov function $V(q) = q$. From the queue update and the fact that $[x]_+ \leq x$,

$$Q_{t+1} = [Q_t - S_t]_+ + A_t \ \leq \ Q_t - S_t + A_t. \tag{58}$$

Therefore

$$V(Q_{t+1}) - V(Q_t) = Q_{t+1} - Q_t \ \leq \ A_t - S_t. \tag{59}$$

Taking conditional expectations given $Q_t = q$ and using Assumptions 3 and 4,

$$\mathbb{E}[V(Q_{t+1}) - V(Q_t) \mid Q_t = q] \leq \mathbb{E}[A_t] - \mathbb{E}[S_t \mid Q_t = q] \tag{60}$$
$$\leq \lambda - \mathbb{E}[S_t \mid Q_t = q]. \tag{61}$$

When $q \geq Q_0$, Assumption 4 yields $\mathbb{E}[S_t \mid Q_t = q] \geq \mu$, and together with the load condition $\lambda < \mu$ we obtain

$$\mathbb{E}[V(Q_{t+1}) - V(Q_t) \mid Q_t = q] \ \leq \ \lambda - \mu = -\delta, \tag{62}$$

where $\delta = \mu - \lambda > 0$. That is, once the queue length exceeds the threshold $Q_0$, the one-step expected drift is uniformly negative.

When $q < Q_0$, we have

$$V(Q_{t+1}) - V(Q_t) \leq A_t, \tag{63}$$

so

$$\mathbb{E}\big[|V(Q_{t+1}) - V(Q_t)| \mid Q_t = q\big] \ \leq \ \mathbb{E}[A_t] + S_{\max} \ \leq \ B \tag{64}$$

for some constant $B$ independent of $q$ (here we used $S_t \leq S_{\max}$ and $\mathbb{E}[A_t^2] < \infty$).

In summary, there exists a nonnegative function $V(q) = q$ whose drift

$$\Delta(q) = \mathbb{E}[V(Q_{t+1}) - V(Q_t) \mid Q_t = q] \tag{65}$$

is strictly negative outside a compact set and is uniformly bounded over the entire state space. By the standard Foster–Lyapunov criterion for Markov chains, $\{Q_t\}$ is positive recurrent, admits a unique invariant distribution, and has a finite first moment in steady state; in particular, there exists $C < \infty$ such that $\mathbb{E}[Q] \leq C$. This proves the proposition. $\square$

### A.2.5 TEMPLATE FOR THE 17-DIMENSIONAL HERA STATE VECTOR

We give a template for the 17-dimensional HERA state vector as a table, with the following fields:

- Name;
- Physical quantity;
- Range;
- Normalization;

Table 1: Template for the 17-dimensional HERA state vector (selected fields).

| Name | Physical quantity | Range | Normalization |
|------|-------------------|-------|---------------|
| gpu_util_inst | Instantaneous GPU compute utilization | $[0, 1]$ | Direct ratio |
| gpu_util_ema | EMA of GPU compute utilization | $[0, 1]$ | Direct ratio |
| vram_util_inst | Instantaneous total VRAM utilization (weights + KV) | $[0, 1]$ | Direct ratio |
| vram_util_ema | EMA of VRAM utilization | $[0, 1]$ | Direct ratio |
| kv_pressure | KV cache occupancy as a fraction of the KV budget | $[0, 1]$ | Direct ratio |
| queue_len | Number of sequence groups in the waiting queue | $[0, Q_{\max}]$ | Divide by $Q_{\max}$ |
| queue_growth | Change in queue length over a recent window | $[-Q_{\max}, Q_{\max}]$ | Divide by $Q_{\max}$ |
| arrival_burst | Coefficient of variation (CV) of arrivals in a recent window | $[0, c_{\max}]$ | Divide by $c_{\max}$ |
| wait_p50 | $p50$ of waiting time for completed requests | $[0, W_{\max}]$ | Divide by $W_{\max}$ |
| wait_p95 | $p95$ of waiting time for completed requests | $[0, W_{\max}]$ | Divide by $W_{\max}$ |
| ttft_p50_ratio | $\log(\text{TTFT}_{p50}/\text{TTFT\_ema}_{p50})$ | $[-R_{\max}, R_{\max}]$ | Divide by $R_{\max}$ |
| ttft_p95_ratio | $\log(\text{TTFT}_{p95}/\text{TTFT\_ema}_{p95})$ | $[-R_{\max}, R_{\max}]$ | Divide by $R_{\max}$ |
| e2e_p50_ratio | $\log(\text{E2E}_{p50}/\text{E2E\_ema}_{p50})$ | $[-R_{\max}, R_{\max}]$ | Divide by $R_{\max}$ |
| e2e_p95_ratio | $\log(\text{E2E}_{p95}/\text{E2E\_ema}_{p95})$ | $[-R_{\max}, R_{\max}]$ | Divide by $R_{\max}$ |
| short_share | Fraction of short-horizon bucket in the current queue | $[0, 1]$ | Direct ratio |
| long_share | Fraction of long-horizon bucket in the current queue | $[0, 1]$ | Direct ratio |
| sla_viol_rate | Fraction of latency-SLO violations in a recent window | $[0, 1]$ | Direct ratio |

