# OpenReview forum: "ATHENA-Serve: An Intelligent Scheduling LLM Serving System via Horizon-Cost Prediction and Hierarchical RL"
_ICLR.cc/2026/Conference — Submitted to ICLR 2026_

### Official Review · Reviewer_N4eY · 2025-10-31

**Soundness:** 3
**Presentation:** 3
**Contribution:** 2
**Rating:** 4
**Confidence:** 2

**Summary:**

The paper proposes ATHENA-Serve, a scheduling framework that 1) predicts each request's horizon (output length) and maps it to KV-cache and compute budgets via ORACLE, and 2) uses HERA, a hierarchical scheduling policy, to make admission, batching, and decode-concurrency decisions within those budgets. On ShareGPT traces with Llama-7B-Chat, it reports up to 1.64x lower latency than baseline serving systems. The central thesis is that budget semantics plus hierarchical control tame head-of-line effects under bursty, heavy-tailed workloads and stabilize tails.

**Strengths:**

1. The budgeted formulation is neat: predicted horizons are converted into calibrated KV/compute budgets, which gives the scheduler a clear, interpretable contract for feasibility checks and homogeneous batching.

2. The hierarchical design decouples global feasibility/admission from local ordering, which reduces search/credit-assignment noise and provides safe guardrails under high load.

3. The evaluation emphasizes tail behavior across multiple arrival regimes and shows consistent left-shifts in p95 TTFT/E2E relative to FCFS-style baselines.

**Weaknesses:**

1. The approach hinges on a trained length predictor and a learned RL policy, but there is no sensitivity analysis for predictor accuracy/placement, policy stability, or contention under different loads/models, which makes robustness hard to judge.

2. The empirical scope is narrow (single model, single GPU, one serving stack version, ShareGPT only), so it is unclear how the method behaves with larger models, multi-GPU/cluster settings, or alternative serving backends.

3. Baselines are limited to FCFS-style systems plus a light "vLLM+Oracle" ablation; stronger SLO-aware or budget-aware schedulers and recent HoL-mitigation systems are not compared, leaving open whether the gains are competitive beyond FCFS.

**Questions:**

1. Can you provide sensitivity curves for (a) horizon prediction error/bias and (b) RL policy noise, showing p50/p95 TTFT and E2E across load, and clarify where the method starts to degrade?

2. How does ATHENA-Serve perform with larger models and multi-GPU (e.g., TP/prefill/decode disaggregation, KV sharding), and does the budget mapping still prevent head-of-line under cross-device effects?

3. Under matched compute and identical traces, how does ATHENA compare against recent SLO-aware/budgeted schedulers and HoL-mitigation systems beyond FCFS-style baselines, and which parts of HERA (admission vs. ordering vs. concurrency) contribute most of the tail gains?

---

### Official Review · Reviewer_rihm · 2025-11-01

**Soundness:** 2
**Presentation:** 3
**Contribution:** 2
**Rating:** 2
**Confidence:** 3

**Summary:**

The paper proposes ATHENA-Serve, a horizon-aware LLM serving scheduler that converts predicted output-length “horizons” into calibrated compute/VRAM budgets (via ORACLE) and uses a hierarchical controller (HERA) to make admission, batching, and concurrency decisions. The key claim is that budgeted, state-aware scheduling mitigates head-of-line blocking and smooths memory peaks, improving p95 latency on ShareGPT traces with Llama-7B on a single A40 GPU.

**Strengths:**

- Addresses on an important problem: tail-latency control and HoL mitigation for LLM serving.
- Interpretable decision making: mapping lengths to budgets makes decisions understandable and deployable.
- Clear intuition: translating noisy length predictions into robust budget classes is sensible and may stabilize decisions under shift.
- Hierarchical control framing makes the policy easier to deploy and reason about than an opaque monolith.

**Weaknesses:**

- All citation formats in the paper are incorrect.
- The motivation is underdeveloped. The paper does not ground the proposed design in concrete SLOs, production traces, or quantified pain points. Without realistic trace analyses (e.g., burst patterns, KV-cache peaks, multi-tenant mix), it’s hard to judge practical necessity over strong heuristics.
- Motivation for hierarchical RL is underdeveloped. The paper does not convincingly show that a learned hierarchical policy is necessary over simpler, robust heuristics (e.g., SJF/SRPT variants with KV-aware caps, max-horizon-per-batch limits, or rule-based admission tuned by load). The current narrative feels like an engineering extension of well-known size-based scheduling with budget guards.
- Scope of evaluation is narrow: single model (Llama-7B-Chat), one GPU class (A40) first-turn ShareGPT prompts only. Lacking multi-turn traces.
- Overhead and complexity are not quantified: added latency from ORACLE inference, telemetry, feasibility checks, and HERA control is not reported. Gains may diminish when accounting for these costs or under lighter loads.

**Questions:**

- Can you provide evidences showing how SJF policies fail due to reponsponse prediction error?
- How are budget class boundaries chosen and calibrated across different models and context lengths? Is there a model-agnostic procedure?
- Can a purely rule-based policy (no RL) with horizon caps, KV ceilings, and age-based priority match your results? Please provide a tuned baseline.
- How sensitive is the policy to length-prediction miscalibration or dataset shift?

---

### Official Review · Reviewer_vRKX · 2025-11-02

**Soundness:** 2
**Presentation:** 3
**Contribution:** 2
**Rating:** 4
**Confidence:** 4

**Summary:**

The paper proposes ATHENA-Serve, a serving scheduler that couples (1) ORACLE, a lightweight output-length predictor that maps each request to one of 8 “horizon” classes and converts that into KV-cache and compute budget constraints, with (2) HERA, a hierarchical RL controller that uses those budgets plus live system signals (utilization, queue state, burstiness) to make admission, batching, and concurrency decisions. The key idea is to avoid brittle shortest-job heuristics by turning noisy length predictions into calibrated resource envelopes that the scheduler enforces to mitigate head-of-line (HoL) blocking in decode. Across regimes, ATHENA reduces p95 latency more than means, it claims up to 1.64× lower p95 latency vs SOTA on ShareGPT.

**Strengths:**

The authors provide clear, closed-form KV/compute budgets and tolerance properties that enable stable feasibility checks and near-homogeneous micro-batches. Leveraging Hierarchical RL that separates admission/envelope from ordering, reduces variance, and enforces safety by construction. The results demonstrate consistent p95 TTFT/E2E improvements across multiple bursty regimes, aligning with the max-horizon argument.

**Weaknesses:**

1. All experiments are single-GPU (A40-48GB), single model (Llama-7B-Chat); no multi-GPU/multi-node results, no interconnect contention, and no MoE or larger models.
2. Comparisons omit several SLO/placement-aware schedulers (e.g., Sarathi-Serve/DistServe split-phase schedulers as configured for SLOs, ExeGPT-style policies, or recent joint placement work). The vLLM+Oracle ablation is helpful but not sufficient.
3. Results focus on means/p95; there is no formal SLO satisfaction analysis (e.g., p99, violation rates, TTFT vs ATGT trade-offs), nor throughput/tokens-per-GPU or cost metrics.
4. ORACLE’s calibration is shown only on ShareGPT-like first-turn prompts; robustness under domain drift is not evaluated.
5. No quantified scheduling overhead at higher request rates; safe-rollback and starvation/fairness properties are argued but not stress-tested at scale.
6. Missing ablations on bin count, tolerance slack, and the contribution of meta-policy vs sub-policy.

**Questions:**

1. The authors need to report p99 TTFT/E2E, violation rates, and joint TTFT/throughput curves for fixed SLOs across the five regimes. How sensitive are results to stricter tails (p99.9)?
2.  How does ATHENA perform with multi-GPU (tensor/pipeline parallel) and multi-node clusters where network/KV paging tails appear? Any data with >1 GPU or 70B-class models?
3. Please add SLO-aware schedulers and split-phase systems configured for SLOs (e.g., Sarathi-Serve, DistServe variants, ExeGPT-like controllers), and include an oracle-length upper bound to isolate policy benefits.
4. How does ORACLE’s calibration hold under topic/domain shifts (e.g., code, long-form writing)? Can you show online recalibration effectiveness and failure modes?
5.  What is the per-tick scheduling latency at 50–200 req/s, and how does tokens/s per GPU change relative to FCFS/SJF?
6. It's interesting to include the study of varying #bins (K), tolerance slack, and confidence-based slack; disable the meta-policy (admission envelope) vs sub-policy (ordering) to quantify each component.
7. Provide waiting-time distribution/Gini or tail fairness metrics to verify that prioritizing short-budget jobs does not starve long ones, especially under the Stress regime.

---

### Official Review · Reviewer_ATYC · 2025-11-03

**Soundness:** 3
**Presentation:** 2
**Contribution:** 2
**Rating:** 4
**Confidence:** 3

**Summary:**

This paper proposes ATHENA-Serve, a deployable, horizon–cost–aware LLM serving scheduler. ATHENA-Serve converts predicted generation horizons into calibrated memory and compute budgets. Rather than forecasting exact trajectories, it senses each request’s compute demands and memory footprint. Guided by these budgeted signals, ATHENA-Serve proactively constrains batching and concurrency to smooth memory peaks, while conditioning scheduling decisions on global system signals.

**Strengths:**

1. User requests scheduling in LLM serving is important.

2. The proposed system ATHENA-Serve proactively constrains batching and concurrency to smooth memory peaks.

3. System experiments demonstrate the performance.

**Weaknesses:**

1. The main concern is the machine learning contribution may not be sufficient. The paper is a system paper on llm serving. Such a system reduces the memory peaks while the model accuracy. The proposed scheduler is like a rule-based policy without learning or reinforcement  learning.

2. The representation learning of such policy is unclear. Ablation study on the hyper parameters can be provided.

3. The convergence or regret analysis on such RL process can be provided.

**Questions:**

1. Figure 1, batch or battch?
2. Line 116, missing reference

---

### Author Response · Authors · 2025-11-28
**Response to Reviewer ATYC**

We thank the reviewer for the thoughtful comments and for pushing us to clarify and strengthen the reinforcement learning and theoretical aspects of our work. In response to your suggestions, we have made several substantial revisions:

*   **Regret and stability analysis added.** We added a new **Appendix A.3** (with subsubsections) that formalizes the meta-policy as an online learning problem over discretized meta-actions. There we prove standard **no-regret guarantees** (via the Hedge algorithm) and a **queue stability result** (via a Foster–Lyapunov drift argument) under budget-constrained admission and concurrency control.
*   **RL nature of HERA clarified.** We revised **Sec. 3.3** and the description around Algorithm 1 to make explicit that HERA is a **hierarchical reinforcement learning policy**, not a fixed rule-based scheduler. The algorithmic “rules” are an executor for a learned meta-policy  $\pi_\phi$ , not hand-tuned heuristics.
*   **Representation learning pipeline made explicit.** We clarified the **two-layer representation learning** pipeline: (i) ORACLE’s prompt-level representation that predicts horizon/length and maps to budgets; and (ii) HERA’s system-level state representation that summarizes 17 telemetry and service-quality features. We added a table in the appendix enumerating all 17 state dimensions, including their physical meaning and normalization.

Below we address your main concerns in detail.

* * *

### (1) “Rule-based” vs. reinforcement learning

**Reviewer’s concern.** You questioned whether our scheduler is essentially a hand-crafted rule-based system rather than a genuine RL-based approach.

**What we changed.**
We now explicitly present HERA as a **parameterized hierarchical RL policy**, and we reorganized the text to clearly separate:

1.  A **learned meta-policy**  $\pi_\phi$  that maps the current system state to a meta-action; and
2.  A **structured executor** (Algorithm 1) that turns this meta-action into concrete scheduling decisions under strict hardware/resource constraints.

Formally, at each decision epoch  $t$ , the scheduler observes a 17-dimensional state vector  $s_t$  (GPU/VRAM utilization, KV pressure, queue statistics, latency statistics, SLO violations, horizon-bucket composition, etc.) and applies a parametric meta-policy

$$
\pi_\phi: s_t \mapsto a_t = (b_t, \theta_t),
$$

where:

*    $b_t$  is a **resource envelope** (admission budget, concurrency limit, KV budget share, etc.), guaranteed to respect hardware constraints;
*    $\theta_t = (\alpha_t, \beta_t, \gamma_t)$  are the **weights** controlling the relative importance of priority, waiting time, and horizon information in the downstream ranking.

Given  $(b_t,\theta_t)$  and ORACLE’s horizon-budget predictions, the executor (Algorithm 1) **filters and ranks** requests:

* it first enforces feasibility with respect to the resource envelope  $b_t$  (no OOM, no violation of KV/compute limits);

* then computes a **linear score**
  $$
  \sigma_i = \alpha_t \cdot \text{priority}_i + \beta_t \cdot w_i + \gamma_t \cdot h_i,
  $$
  and selects the feasible set with the highest scores.

Crucially, this executor is **not the policy**; it is a deterministic decoder for the meta-action produced by  $\pi_\phi$ . The actual scheduling behavior is driven by the parameters  $\phi$ , which are learned via RL from a shaped reward that balances throughput, tail TTFT, completion rate, and queue penalties. The policy is trained via interaction with a simulated environment, with a curriculum of arrival patterns (from near-steady traffic to heavy-burst/OOM-prone regimes), rather than via manual tuning.

To make this explicit in the paper, we have:

*   rewritten the description in **Sec. 3.3** to emphasize that HERA is a **hierarchical RL policy**;
*   renamed Algorithm 1 as an “executor given (admission envelope, weights)” to avoid the impression of a fixed heuristic;
*   clarified that the “rules” encode **safety and interpretability constraints**, while the high-level decision  $(b_t,\theta_t)$  is entirely learned.

We believe these clarifications show that our design is not a rule-based heuristic, but a structured RL approach where the “rules” only serve to safely decode the learned meta-actions.

---

### Author Response · Authors · 2025-11-28
**Response to Reviewer ATYC**

### (2) Representation learning in the policy

**Reviewer’s concern.** You noted that the representation learning in our policy was not clearly explained, making it hard to see where learning actually occurs and how it leverages the rich information in prompts and system telemetry.

**What we changed.**
We now explicitly describe a **two-stage representation learning pipeline**:

1. **Request-level representation via ORACLE.**
   ORACLE takes a user prompt as input and passes it through a distilled LLM encoder to obtain a prompt embedding. From this embedding, it jointly predicts:

   *   a **horizon class** (e.g., instant / short / medium / long / extreme) indicating expected decoding length;
   *   a **continuous length estimate**  $\hat L$ ;
   *   a **confidence / calibration score**.
       These heads are trained together using cross-entropy, regression losses, focal and adjacency regularization, long-tail reweighting, and curriculum-driven terms. The goal is to enforce:
   *   **smoothness** across neighboring horizon buckets;
   *   robustness on long-tail prompts;
   *   calibration of length predictions.
       The outputs  $(\hat c, \hat L, \hat p)$  are then mapped via analytic formulas to **KV and compute budgets** such as  $B_{\mathrm{KV}}(P,\hat L)$  and  $F_{\mathrm{dec}}(P,\hat L)$ . This mapping creates a **horizon-to-budget representation** that encodes the resource implications of each prompt.

2. **System-level representation via HERA’s state vector.**
   On top of these request-level horizon-budget signals, HERA constructs a 17-dimensional **system state vector**  $s_t$  summarizing:

   *   hardware utilization (instantaneous and EMA GPU/VRAM utilization, KV-cache pressure);
   *   queue structure (queue length, queue growth over a window, per-bucket composition, arrival burstiness);
   *   user-facing latencies (TTFT/E2E p50/p95 vs EMA baselines);
   *   SLA quality (recent SLO violation rate).
       Each component is carefully normalized (e.g., by maximum queue length or log-ratio against EMA baselines) and in many cases smoothed via EMAs with different windows. This state vector is fed into a parametric mapping

   $$
   f_\phi: s_t \to (b_t,\theta_t),
   $$

   implemented as a lightweight neural network. During RL training,  $f_\phi$  learns a **representation of system regimes** (steady state, transient bursts, high KV pressure, prolonged backlog, etc.) and maps them to different resource envelopes and scoring weights.
   In the revision, we have added an **appendix table** that enumerates all 17 state dimensions with four key fields:

   *   Name;
   *   Physical quantity;
   *   Range;
   *   Normalization.
       This makes the system-level representation explicit and easy to reproduce.

Conceptually, the linear scoring rule in Algorithm 1 is just a transparent final decoder that converts the meta-action  $\theta_t$  into request scores, while the **expressive representation learning** happens in:

*   the learned prompt representation in ORACLE; and
*   the learned mapping from the 17-dimensional state representation to  $(b_t,\theta_t)$  in HERA.

We highlight this two-stage representation-learning structure in the revised sections so readers can clearly see how the model leverages both prompt-level and system-level information.

---

### Author Response · Authors · 2025-11-28
**Response to Reviewer ATYC**

### (3) Missing regret and stability analysis

**Reviewer’s concern.** You asked for a more rigorous regret-style analysis and a discussion of stability guarantees for the proposed RL-based scheduler.

**What we changed.**
We have added a new **Appendix A.3** that gives a **formal analysis** of regret and stability for the meta-policy under mild assumptions compatible with our implementation.

1. **No-regret guarantee over discretized meta-actions.**
   We model the meta-action  $a_t = (b_t,\theta_t)$  as living in a compact continuous space  $\mathcal{A}\subset\mathbb{R}^d$ . We then introduce a finite  $\varepsilon$ \-grid  $\widetilde{\mathcal{A}}$  over  $\mathcal{A}$ , and assume:

   *   rewards are bounded,  $r_t(a)\in[0,1]$ ;
   *   rewards are **Lipschitz in the action**,  $|r_t(a)-r_t(a')|\le L\|a-a'\|_2$ , which follows from the smoothness of ORACLE’s budget mappings and the smooth bounded reward shaping we use.
       Over the finite set  $\widetilde{\mathcal{A}}$ , we consider a standard **exponential-weights (Hedge)** algorithm on meta-actions. We prove that Hedge enjoys the usual **no-regret bound**:

   $$
   \mathbb{E}\Big[\sum_{t=1}^T (r_t(a^\star)-r_t(a_t))\Big] \le \sqrt{2T\log K},
   $$

   where  $a^\star$  is the best fixed meta-action in hindsight in  $\widetilde{\mathcal{A}}$  and  $K = |\widetilde{\mathcal{A}}|$ . Thus, the average regret of the meta-policy relative to the best discrete meta-action vanishes as  $T\to\infty$ .

2. **Approximation error between continuous and discrete optima.**
   Using the Lipschitz property, we show that if  $\widetilde{\mathcal{A}}$  is an  $\varepsilon$ \-grid of  $\mathcal{A}$ , then the total reward of the best continuous meta-action  $a^\star_{\mathrm{cont}}$  is at most  $L\varepsilon T$  larger than that of the best discrete grid meta-action  $a^\star_{\mathrm{disc}}$ :
   $$
   \sum_{t=1}^T r_t(a^\star_{\mathrm{cont}}) \le \sum_{t=1}^T r_t(a^\star_{\mathrm{disc}}) + L\varepsilon T.
   $$
   Combining this with the Hedge bound yields a regret guarantee **relative to the continuous optimum**:
   $$
   \mathbb{E}\Big[\sum_{t=1}^T (r_t(a^\star_{\mathrm{cont}})-r_t(a_t))\Big] \le \sqrt{2T\log K} + L\varepsilon T.
   $$
   For sufficiently fine discretization ( $\varepsilon$  small), the additional term is negligible, showing that the learned meta-policy is essentially **no-regret w.r.t. the best continuous meta-action**.

3. **Queue stability under budget-constrained scheduling.**
   We also add a **queueing-theoretic stability analysis** for the backlog process. Modeling the total backlog  $Q_t$  via:
   $$
   Q_{t+1} = [Q_t - S_t]_+ + A_t,
   $$
   where  $A_t$  is the incoming work and  $S_t$  is the completed work determined by our budget-constrained policy, we assume:

   *   i.i.d. arrivals with mean  $\lambda$  and finite second moment;
   *   a **capacity upper bound**  $S_t \le S_{\max}$  induced by hardware limits;
   *   a **non-idling condition**: when  $Q_t$  is large enough, the expected service rate  $\mathbb{E}[S_t\mid Q_t=q]\ge \mu$  for some  $\mu>0$ ;
   *   the **subcritical load** condition  $\lambda < \mu$ .
       Using the Lyapunov function  $V(q)=q$  and a standard Foster–Lyapunov drift argument, we show that the backlog process  $\{Q_t\}$  is **positive recurrent** and has a finite steady-state expectation, i.e., the queue is stable whenever the long-term arrival rate is below the effective service capacity implied by the resource envelope. This formalizes the intuition that HERA’s budget-constrained control prevents the system from diverging under realistic loads.

These additions provide the regret-style and stability guarantees you requested, grounded in standard online learning and queueing theory, and rigorously justify the behavior we observe empirically.

* * *

We hope that these clarifications and the new appendix address your concerns about:
(i) whether the policy is genuinely RL-based rather than rule-based;
(ii) where and how representation learning occurs; and
(iii) what theoretical guarantees we can offer on regret and stability.
We are grateful for your feedback, which has significantly improved both the clarity and the rigor of the paper.

---

### Author Response · Authors · 2025-11-28
**Response to Reviewer rihm**

We sincerely appreciate the reviewer’s careful reading of our paper and the thoughtful, constructive feedback. Your comments have been very helpful in clarifying the motivation of our work, making our experimental scope more transparent, and sharpening the description of our horizon-aware budget design.

* * *

**On “How are budget class boundaries chosen and calibrated? Is there a model-agnostic procedure?”**

Here we provide a precise description of how we construct and calibrate the budget (horizon) classes, in a way that is shared across models and context lengths.

1. **Normalized cost proxy (model-agnostic form).**
   For each decoder-style model  $m$ , with KV capacity  $M_{\text{KV}}^{(m)}$  and decode capacity  $C_{\text{dec}}^{(m)}$ , and for a request with prompt length  $P$  and output length  $L$ , we have analytic formulas for KV and decode budgets:
   $$
   \mathbf{B}^{(m)}(P,L) = \big(B_{\text{KV}}^{(m)}(P,L),\; F_{\text{dec}}^{(m)}(P,L)\big).
   $$
   We combine them into a **normalized scalar cost**:
   $$
   C^{(m)}(P,L) = \lambda_{\text{KV}} \cdot \frac{B_{\text{KV}}^{(m)}(P,L)}{M_{\text{KV}}^{(m)}} + \lambda_{\text{F}} \cdot \frac{F_{\text{dec}}^{(m)}(P,L)}{C_{\text{dec}}^{(m)}},
   $$
   where  $\lambda_{\text{KV}},\lambda_{\text{F}}>0$  are fixed global weights. For a given model and prompt length, this cost is **monotone in  $L$ **, and the functional form is the same across models; the only model-specific parts are the capacity constants.

2. **Defining budget classes by quantiles in cost space.**
Given a log of  $(P_i,L_i)$  pairs for model  $m$  under a particular context configuration, we compute the corresponding cost samples  $C^{(m)}(P_i,L_i)$  and their empirical distribution  $\hat{\mu}^{(m)}$ . For a chosen number of classes  $K$ , we define quantile boundaries:
$$
c_k^{(m)} = \text{Quantile}_{\hat{\mu}^{(m)}}\!\left(\frac{k}{K}\right)
$$
 The  k -th budget class is:
 $$
C_k^{(m)} = \{(P,L): C^{(m)}(P,L) \in {I}_k^{(m)}\}.
$$
In words, each class corresponds to a band of normalized resource cost, so the classes have a similar “cost meaning” across models and context lengths even though the raw lengths differ.
3. **Representative lengths vs. online budgets.**
   For interpretability, we can associate each class with a representative length when describing the horizons. However, online budgeting always uses the continuous predicted length  $\hat L$  (plus slack) inserted into the analytic formulas  $B_{\text{KV}}^{(m)}(P,\hat L)$  and  $F_{\text{dec}}^{(m)}(P,\hat L)$ . We do _not_ snap back to the representative length at runtime. This keeps the mapping smooth while the class boundaries themselves are derived in a unified way for any model/context by re-running the same quantile procedure on its logs.

In this sense, the **procedure is model-agnostic**: the form of the cost mapping and the quantile-based partitioning are identical across models and context lengths; the only things that change are the capacity constants and the empirical distribution used to set the actual numeric boundaries.

---

**Citation format.**
We took this comment very seriously and carefully re-checked all references and in-text citations against the official conference style guidelines. In doing so, we did not find a systematic mismatch in the citation scheme itself (e.g., author–year vs. numeric), but we did cleaned up a few typos and notation inconsistencies in the text. If there are specific examples where our formatting still deviates from the intended style, we would be very grateful to correct them in the camera-ready version.

---

### Meta-Review · Area_Chair_Dhkh · 2026-01-06

**Summary:**

The reviewers agree that the paper addresses an important and practical problem in LLM serving, namely tail-latency control under bursty and heavy-tailed workloads, and they appreciate the interpretable budget-based formulation and the hierarchical scheduling design of ATHENA-Serve. The system demonstrates consistent p95 latency improvements over FCFS-style baselines on ShareGPT traces, and the idea of mapping noisy length predictions to calibrated resource budgets is viewed as intuitive and potentially useful. However, multiple reviewers raised concerns that the contribution is primarily systems-engineering oriented with limited machine learning novelty, and that the empirical evaluation is too narrow to support strong claims about scalability.

**Reviewer Concerns:**

While the rebuttal clarified the budget calibration procedure and addressed some presentation issues, major concerns remain regarding the lack of multi-GPU or multi-node experiments, limited baseline comparisons to state-of-the-art SLO-aware schedulers, missing p99 or violation-rate analyses, unquantified scheduling overheads, and insufficient evidence that the hierarchical RL components provide clear advantages over simpler, well-tuned heuristic or rule-based policies.

**Reviewer Scores:**

Reviewer scores would likely remain unchanged after discussion.

---

### Decision · Program_Chairs · 2026-01-26

Reject